# Cadaveric simulation versus standard training for postgraduate trauma and orthopaedic surgical trainees: protocol for the CAD:TRAUMA study multicentre randomised controlled educational trial

Hannah K James ![ORCID],[1,2] Giles T R Pattison,[2] Joanne D Fisher,[1] Damian Griffin[1,2]

¹Clinical Trials Unit, University of Warwick, Warwick Medical School, Coventry, UK
²Trauma & Orthopaedic Surgery, University Hospitals Coventry and Warwickshire NHS Trust, Coventry, UK

**Correspondence to**
Dr Hannah K James;
h.smith.1@warwick.ac.uk

## ABSTRACT

**Introduction** The quantity and quality of surgical training in the UK has been negatively affected by reduced working hours and National Health Service (NHS) financial pressures. Traditionally surgical training has occurred by the master-apprentice model involving a process of graduated responsibility, but a modern alternative is to use simulation for the early stages of training. It is not known if simulation training for junior trainees can safeguard patients and improve clinical outcomes. This paper details the protocol for a multicentre randomised controlled educational trial of a cadaveric simulation training intervention versus standard training for junior postgraduate orthopaedic surgeons-in-training. This is the first study to assess the effect of cadaveric simulation training for open surgery on patient outcome. The feasibility of delivering cadaveric training, use of radiographic and clinical outcome measures to assess impact and the challenges of upscaling provision will be explored.

**Methods and analysis** We will recruit postgraduate orthopaedic surgeons-in-training in the first 3 years (of 8) of the specialist training programme. Participants will be block randomised and allocated to either cadaveric simulation or standard 'on-the-job' training, learning three common orthopaedic procedures, each of which is a substudy within the trial. The procedures are (1) dynamic hip screw, (2) hemiarthroplasty and (3) ankle fracture fixation. These procedures have been selected as they are very common procedures which are routinely performed by junior surgeons-in-training. A pragmatic approach to sample size is taken in lieu of a formal power calculation as this is novel exploratory work with no a priori estimate of effect size to reference. The primary outcome measure is the technical success of the surgery performed on patients by the participating surgeons-in-training during the follow-up period for the three substudy procedures, as measured by the implant position on the postoperative radiograph. The secondary outcome measures are procedure time, postoperative complication rate and patient health state at 4 months postoperation (EQ-5D—substudies 1 and 2 only).

## Strengths and limitations of this study

► This is the first randomised controlled trial assessing the impact of cadaveric simulation training on clinical outcomes.
► Patient-centred outcome measures are used to measure an educational intervention for surgeons.
► Multicentre study to maximise external validity of the results.
► The training dose is small as cadaveric training is expense to deliver.
► Pragmatic approach to sample size which is limited by the capacity of the surgical training centre.

**Ethics, registration and dissemination** National research ethics approval was granted for this study by the NHS Research Authority South Birmingham Research Ethics Committee (15/WM/0464). Confidentiality Advisory Group approval was granted for accessing radiographic and outcome data without patient consent on 27 February 2017 (16/CAG/0125). The results of this trial will be submitted to a peer-reviewed journal and will inform educational and clinical practice.

**Trial registration number** ISRCTN20431944

## INTRODUCTION

It is imperative that surgeons are trained to a high standard, so they can perform safe and effective operations for patients. The quality and quantity of surgical training in the UK is currently under threat from a 'perfect storm' of factors.[1] These include reduced working hours,[2 3] shift-based working patterns[4] with the loss of the traditional surgical firm and a move to expedite training and shorten specialist programmes.[5 6] This is set within a climate of unprecedented financial austerity in the NHS and ever-increasing service pressures.

Simulation offers a solution to some of these challenges by moving the early part of the surgical learning curve away from patients into a controlled environment,[7] where skills may be more rapidly acquired as compared with the clinical environment. Simulation is also potentially a very efficient way of training, as large numbers of trainees can be trained simultaneously, at an intensity not feasible in the clinical environment due to competing service demands.

Cadaveric simulation—training using deceased, preserved or fresh human bodies—is a particularly promising modality for training. Fresh-frozen cadavers retain many of the soft tissue handling characteristics seen in live patients, and in combination with presenting the correct anatomy, particularly complex neurovascular relationships, may offer a more realistic simulated operation than would be possible on a plastic model or virtual reality simulator.[8 9] Cadaveric material does not bleed[10] and hence may be less useful for simulating procedures where haemorrhage control is an important feature.

The operating theatre environment can be simulated, including (but not limited to) surgical dress, draping, instrumentation and multidisciplinary team. This 'whole dress rehearsal' for surgery may enhance development of non-technical skills in addition to the technical operative surgical skills.[11]

There are several challenges in delivering cadaveric simulation training. It is expensive to provide,[9] particularly when cadaveric material has to be purchased under license, where there is not a local body donation programme. It requires considerable infrastructure to deliver, including specialist wet laboratory facilities with the appropriately trained staff. These challenges become particularly pressing when provision of cadaveric training on a large scale is considered, and are an important driver in the development of high-quality evidence of educational impact. This evidence is necessary before considerable financial investment can be recommended in providing cadaveric simulation training on a larger scale.

There is abundant low-quality evidence showing cadaveric simulation may induce short-term skill improvement as measured by subjective and behavioural metrics, but there is a lack of high-quality, quantitative evidence that skills learnt in cadaveric simulation can transfer to the workplace, leading to improved outcomes for patients.[8]

Our trial attempts to address this evidence deficit, and is both topical and timely.

## GOOD CLINICAL PRACTICE
This trial will be undertaken in compliance with Good Practice Guidelines, complying with the Declaration of Helsinki and UK Legislation. Warwick standard operating procedures (SOPs) will be followed.

## CONSOLIDATED STANDARDS OF REPORTING TRIALS
The results of the trial will be reported in line with the Consolidated Standards of Reporting Trials (CONSORT) statement.[12] This protocol has been written according to the Standard Protocol Items: Recommendations for Interventional Trials reporting guidelines.[13]

## AIM
The aim is to determine which of the two surgical training strategies for junior orthopaedic surgeons-in-training lead to the best patient outcomes for three common procedures.

## OBJECTIVES
1. To assess the impact of a cadaveric simulation training intervention on the patient outcome of operations performed by junior orthopaedic surgeons-in-training.
2. To define the early learning curve of dynamic hip screw (DHS), hemiarthroplasty and ankle fracture fixation.
3. To explore the feasibility of using postoperative X-rays to assess technical skill.

## METHODS AND ANALYSIS
### Study design
This is a UK multicentre, two-arm, group parallel randomised controlled educational trial.

### Sample size
This trial is the first attempt to objectively measure transfer of open operative skills from cadaveric simulation into the workplace using patient-based outcome measures. There is no available estimate of effect size to reference against a priori in determining sample size, therefore a pragmatic approach to sample size will be taken in lieu of a formal power calculation. The surgical training centre can accommodate 16 delegates at one time and financial resources permitted one iteration of the cadaveric training course. Our maximum sample size is therefore 16 participants in each arm of the study.

## OUTCOME MEASURES
### Radiographic outcomes
The radiographs will be obtained electronically from hospital servers and the implant position measured manually using computer software. The operations will be identified retrospectively by access to the participating surgeons' electronic logbooks. The measurements vary by operation type and are defined as follows.

### Substudy 1: DHS
1. Primary outcome
   i. Tip–apex distance (mm).
2. Secondary outcomes (in order of importance)
   i. Lag screw position in the femoral head (defined by Cleveland Zones).
   ii. Plate flush to lateral femoral cortex (binary Y/N).
   iii. Eight cortex hold for plate screws (binary Y/N).

## Substudy 2: hemiarthroplasty

1. Primary outcome
   i. Leg length discrepancy (mm).
2. Secondary outcomes (in order of importance).
   i. Femoral stem alignment (degrees off neutral).
   ii. Cement mantle quality (Barrack grade score).
   iii. Femoral offset change relative to native hip (mm).

## Substudy 3: ankle fracture fixation

1. Primary outcome
   i. Medial clear space (mm).
2. Secondary outcomes
   i. Lateral malleolar displacement (mm).
   ii. Tibiofibular clear space (mm).
   iii. Talocrural angle (degrees).
   iv. Medial malleolar displacement (mm).

## Clinical outcomes

The clinical outcome measures for substudies 1–3 are as follows:

1. Procedure time
   Defined as knife-to-skin/surgical start time to wound closure/surgical stop time. These will be obtained from hospital theatre management systems. Procedure time has been chosen as an outcome measure as there is evidence in the literature that procedure time is inversely related to experience, and so can be used as a surrogate measure of technical proficiency.[14]
2. Intraoperative radiation dose to patient
   Defined as time under fluoroscopy (seconds) and radiation dose ($mGym^2$). There is evidence that with increasing experience and skill, surgeons use less intraoperative X-rays to adjust the position of the fracture and implant.[14] Hemiarthroplasty does not require fluoroscopy so this will not be used as an outcome measure for substudy 2.
3. Postoperative complication rate
   The complications of interest are the acute postoperative complications during the inpatient admission. These will be subcategorised as acute medical complications (hospital-acquired pneumonia, renal complications, cardiac complications, Deep Vein Thrombosis (DVT)/Pulmonary Embolism (PE) and surgical complications (wound infection, wound dehiscence, metalwork failure, deep infection).
4. Health state at 4 months postoperation (EQ-5D).
   Health state at 4 months postoperation will be measured using EQ-5D, which is a standardised instrument measuring generic health status, which has been widely validated in clinical trials. These data are being collected separately as part of the WHiTE comprehensive cohort study of patients with hip fracture (ISRCTN63982700) and reported elsewhere.[15] EQ-5D will be used for substudies 1 and 2 only as these involve hip fractures.

## Screening and eligibility

Orthopaedic surgeons-in-training in their first, second or third specialist training year in the West Midlands Workforce Deanery will be eligible to participate in the trial. Eligible trainees will be identified by liaison with the training programme directors for Trauma & Orthopaedic surgery in the three West Midlands schools; Warwick, Birmingham and Oswestry. An invitation email will be sent to all eligible trainees by programme administrators at the deanery.

### Inclusion criteria

1. Trauma & Orthopaedic surgeon-in-training in West Midlands school (Warwick/Birmingham/Oswestry).
2. In specialty training years 1–3.
3. Willing and able to attend a 2-day cadaveric simulation training course at the West Midlands Surgical Training Centre (WMSTC), Coventry.

### Exclusion criteria

1. Unavailable on course dates.

## Consent
### Surgeon participants

Potential study participants will be provided with written and verbal information about the study. Consent will be obtained by the trial team. The right to refuse participation without giving reasons will be fully respected, and enrolled participants will be free to withdraw from the study at any time without reason, and without prejudice to their training. All participants will be provided with the contact information of a team member who can provide further information about the study. All participants who are allocated to the control group will have the opportunity to undertake the cadaveric simulation training intervention at the end of the study follow-up. This provision is being offered so that the control group are not disadvantaged in their access to educational opportunity by virtue of being randomised to the control group.

### Patients whose operations are assessed

Patients who undergo an operation by a surgeon who is participating in the study will not be separately consented to allow access to radiographs to assess their implant position or clinical outcome data. Permission to access this information for the purposes of this study without patient consent has been granted from the confidentiality advisory group (16/CAG/0125). It is recognised that seeking consent from a group of primarily elderly, frail patients to assess low risk, routine clinical data in a secure manner for a trial they are not directly participating in would be unduly burdensome for the patients. All patient data will be fully anonymised and handled securely in line with university data regulations.

## Randomisation

Participants will be randomised at the point of recruitment using block randomisation (block size 4) to generate a random sequence list, to which participants

will be assigned in the order that they enter the study. The allocation sequence will be generated by a senior medical statistician, participants will be enrolled by the trial team.

### Postrandomisation withdrawals
Withdrawn participants will not be replaced.

### Study setting
The study participants will be on training rotations within the regional hospitals of the West Midlands during the study follow-up. The hospitals where trainees have been working, and performing operations, during the study follow-up will be identified from the participants electronic surgical logbook records.

### Interventions
#### Control group
The control group will undertake standard residency training according to the master-apprentice model, which is the current standard practice in UK. No additional training or access to learning materials will be provided beyond the fortnightly didactic teaching sessions which are delivered as a part of routine training.

#### Intervention group (cadaveric simulation trained)
Participants allocated to the intervention group will receive an intensive, 2-day cadaveric simulation training course at the start of the training year, where four common orthopaedic surgical procedures will be taught (DHS, hemiarthroplasty, ankle fracture fixation and lower limb fasciotomy). All intervention participants will receive training on all four procedures, which will be considered separately in the analysis as individual substudies (as they have different radiographic outcome measures). The fasciotomy procedure is included as a 'filler' to make the course structure work, and chosen because it is an important high-stakes, anatomically critical operation that is rarely performed by trainees. Outcomes related to the fasciotomy procedure will not be collected or included in the analysis.

#### The cadaveric simulation training course
The course will be delivered in September at the start of the surgical training year (which runs August to August). The course will take place in the WMSTC at the University Hospital Coventry & Warwickshire (UHCW). The WMSTC is a specialised wet-laboratory facility for delivering cadaveric training, and has an experienced dedicated faculty to facilitate training delivery.

The course will consist of two full days of teaching, with expert consultant faculty teaching on fresh-frozen hemi-cadavers (waist-to-toe-tip). The participant:faculty ratio will be 2:1, and participant:cadaver ratio will be 2:1. Each participant will undertake each of the four procedures in their entirety as primary surgeon ('skin-to-skin'), and will act as assistant when their partner is the primary surgeon four times. Hence each participant is exposed to eight procedures during the course.

The environment and psychological fidelity of the simulation will be maximised by providing:
1. Full surgical dress including masks, gloves, gowns and lead X-ray aprons.
2. The usual disposable surgical drapes.
3. Skin preparation (iodine solution) to prepare the surgical site, and participants and faculty will be asked to observe the usual sterile field precautions as in real theatre.
4. Full surgical instrument trays, surgical implants and cement (for hemiarthroplasty) of the same type as in real theatre will be used.
5. Image intensifier (mobile X-ray) will be available for intraoperative use.
6. Background noise levels and room temperature were maintained at what would usually be expected in the operating theatre.

The simulated operating theatres will be set up within the WMSTC as two parallel round-robin circuits. The two stations requiring X-ray use (DHS and ankle open reduction internal fixation (ORIF)) will be set up at the far end of the room to create a radiation zone and where appropriate, standard precautions will be used. Careful consideration will be given to the optimum sequential use of the cadaveric specimens in planning the course structure. For example, it is necessary that the DHS station precedes the hemiarthroplasty station as it would obviously not be possible to perform a DHS operation when the femoral head had been removed. Similarly, the fasciotomy incisions would compromise the soft tissue envelope of the lower limb to a sufficient degree that the fidelity of the ankle ORIF station would be compromised. It is important to make the best and most efficient use of the cadaveric material, for both ethical and financial reasons.

### Blinding
The participants cannot be blinded to the type of training they receive, neither can the trial team in organising the cadaveric simulation training. The trial team will take no part in the training of participants. The assessment of radiographic images will be made blinded to group allocation.

### Adverse event management
In the unlikely event of a serious adverse event, the chief investigator will report to the sponsor (University of Warwick), ethics committee and project supervisors.

### Patient and public involvement
There was no direct patient or public involvement in the design of the study, although clearly training competent surgeons is in the public interest. There is qualitative work to be done around this trial to better understand patient expectations of surgical training.

### End of trial
The trial will end when all the radiographic and clinical outcome data have been collected from the participating sites. The trial will be stopped prematurely if required by

the ethics committee, following recommendations from the sponsor, or if funding for the study is withdrawn. The research ethics committee and confidentiality advisory group will be notified in writing once the trial is complete.

## Trial oversight

This trial is being undertaken as part of a doctoral research project (HKJ), and supervised by three senior supervisors (DG, JDF, GTRP). The supervisors will act as the trial management group and steering committee. The trial is being conducted within a registered Clinical Trials Unit (CTU), and will follow the CTU SOPs.

## Data collection plan

Data on the numbers of procedures performed by the participating surgeons at baseline will be collected. The operations performed by the participants during study follow-up will be identified by the surgeons' electronic logbook. Only procedures coded as 'S-TS: supervised-trainer scrubbed' or 'S-TU: supervised trainer unscrubbed' will be included in the analysis. This is to ensure that only procedures where the trainee has performed the key parts (S-TS) or the entire procedure (S-TU) are included. If further information on supervisor input/takeover is required this can be obtained by accessing the corresponding procedure based assessment (PBA) record for the operation. PBAs are routinely collected as part of training.

Procedure data will be extracted and anonymised to study identifier by the electronic logbook data team, before being sent to the trial team. The data will include operation type, date, hospital, hospital ID, patient age, American Society of Anaesthesiologists Grade and supervision code. The radiographs and clinical outcome data relating to these procedures will then be obtained from the study sites via liaison with the respective Research & Development Departments. Data will be entered into a secure trial database on a professionally encrypted trial-specific computer, fully anonymised with only study identifiers. Once data collection is complete, and prior to analysis, range checks for data values will be undertaken, and data will be double checked on entry to the statistical software package. The project supervisors will act as the data monitoring committee. No interim analysis will be undertaken. The trial team and statistician will have access to the final trial dataset.

## Statistical analysis plan

Baseline data including completed months of training and number of prior procedures performed will be summarised and compared between the two arms of the study. A CONSORT chart showing the flow of participants through the study will be produced. The three taught procedures (substudies 1–3) will be analysed and reported individually.

The main analysis will investigate and report differences between the two groups with respect to the implant positions (as measured from radiographs), the procedure

times, the intraoperative radiation dose to the patient, and patient outcomes, as measured by postoperative complications and health state at 4 months postoperation (hip fractures only).

Statistical tests will be two-sided and considered to demonstrate a significant difference when $p < 0.05$. Temporal trends by group for implant position, procedure time and radiation dose will be presented. Linear mixed-effects models will be fitted to allow for within-surgeon correlation between repeated observations (surgeon clustering as a random effect), and to adjust for important covariates such as patient condition, age and surgeon experience. These will be summarised by plotting individual learning curves, and then modelled to estimate the overall learning curves for the two arms of the study.

Descriptive statistical analyses of between-group comparisons will be presented for complication rate and health state, with temporal analysis of the latter being reported if appropriate and feasible. The statistical analysis will be supervised and checked by a senior medical statistician at Warwick University.

In the event of missing data, statistician advice will be sought on multiple imputation.

## ETHICS AND DISSEMINATION

Master-apprentice 'on-the-job' training for surgeons is the current training standard in the UK,[10 16] and therefore the control arm of the study reflects usual practice. The cadaveric simulation training intervention is an experimental educational intervention and does not expose trial participants to any substantial risks of harm. The trial results will be reported in accordance with the CONSORT statement, and disseminated through publication in peer-reviewed journals and conferences. The results of the trial will be presented to Health Education England and the Royal Surgical College. The dataset, statistical code and technical appendices will be made available on request to the corresponding author. The study was approved by the NHS Research Authority South Birmingham Research Ethics Committee (15/WM/0464).

**Contributors** HKJ designed the study and wrote the manuscript. GTRP codesigned the study and the intervention and edited the manuscript. JDF edited the manuscript, made a substantial contribution to the design and is lead supervisor for the qualitative part of the project. DG codesigned the study, edited draft protocols and is lead supervisor for the quantitative part of the project.

**Funding** This work was supported by Versus Arthritis grant number 20845.

**Competing interests** None declared.

**Patient consent for publication** Not required.

**Provenance and peer review** Not commissioned; externally peer reviewed.

**ORCID iD**
Hannah K James http://orcid.org/0000-0002-0535-3062

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
