## [Reviewer comments · BMJ Open]

ARTICLE DETAILS

TITLE (PROVISIONAL)	Cadaveric simulation vs standard training for postgraduate trauma & orthopaedic surgical trainees: protocol for the CAD:TRAUMA study multi-centre randomised controlled educational trial
AUTHORS	James, Hannah K; Pattison, Giles; Fisher, Joanne; Griffin, Damian

VERSION 1 – REVIEW

REVIEWER	Amandus Gustafsson Copenhagen Academy for Medical Education and Simulation
REVIEW RETURNED	24-Mar-2020

GENERAL COMMENTS	Thank you for the opportunity to review your manuscript. Investigating evidence of transfer of learning from the simulated setting to the clinical setting is important. Especially since the training modality of cadavers has been around for centuries, but as the authors point out, only have been investigated to a small degree compared with more novel modalities as virtual reality and computer assisted simulators. I find the protocol interesting, but have quite serious concerns chiefly between planned sample and outcomes. Though there is several meta-analysis available on simulation-based training and the effect on patient outcomes that a sample size calculation could be based on, I agree with the authors that is reasonable with a pragmatic approach as training modality and outcomes vary hugely from previously published studies and this study. Also, the number of participants is aligned with similar studies. However, as the proposed study is planed, I'm seriously concerned for a type 2 error. The two main issues are 1) supervisor bias and 2) insufficient dose/exposure. When conduction transfer studies from a simulated setting to the clinical setting for procedures of high complexity an invariable obstacle is the bias of the supervising surgeon as it would be hugely unethical to let novice surgeons operate on their own. We must assume that the senior surgeon will correct the trainee or even temporarily take over the operation to assure an optimal surgical result. This will in my opinion to quite a high degree reduce the effect, if any. This is true of both radiological outcome, but especially so for patient outcomes. Your dose, in this case cadaver procedure training, is very small. You plan to let the intervention group train each procedure only once. Though this intervention may have an effect, I find it very unlikely that you will detect it with your sample size. To illustrate, I have recently published a study (Gustafsson A, Pedersen P, Rømer TB, Viberg B, Palm H, Konge L. Hip-fracture
--

	osteosynthesis training: exploring learning curves and setting proficiency standards. Acta Orthop 2019; 90(4): 348-53) on simulation-based training. Among others, we found that the median amounts of attempts the novices need to reach their learning plateau was 8 (range: 4-18). Also, your subjects are quite heterogeneous. You state that you will include trainees in their specialist training year 1 through 3. This is a huge difference in baseline competency and I would fear that the effect of the first year trainees will be washed out or be diminished by the third year trainees. It's not stated, but at the very least you will have to look into this problem during randomization as a skewness between the groups will seriously harm the interpretation of results. Based on your available sample size I would suggest that you readjust your focus from radiological and clinical outcomes measures to trainee behavioural outcomes with validated assessment tools, if you choose to continue to address comparing training to no training. On the above points I would suggest you read Cook DA, Hatala R. Got power? A systematic review of sample size adequacy in health professions education research. Adv in Health Sci Educ 2015; 20: 73-83 and Cook DA, West CP. Reconsidering the Focus on "Outcomes Research" in Medical Education: A Cautionary Note. Acad Med 2013; 88: 162-67. Specific comments: Page 7, line 52-54: "where skills can be rapidly acquired" – do you have any evidence that skills can be acquired especially fast in a simulated setting as supposed to clinical practise? Page 7, line 54: "competency can be assessed before trainees are released into clinical practice." as I understand it, the study does not aim to assess the trainee's competencies in the simulated setting. Page 7, line 56: "Simulation is also potentially a very effective way of training" I disagree. There's ample evidence that simulation-based training of procedural skills is the superior training modality. Page 20, line 18: "The trial will end when data collection is complete." I cannot find any information in the manuscript as to when data collection is complete. Please state. Page 20, line 47: "Data collection plan" If possible, it would certainly improve the confidence of your results, if you could report how many of the investigated procedures the trainees had performed prior to study enrolment.
--	--

REVIEWER	Don Anderson Department of Orthopedics and Rehabilitation University of Iowa Carver College of Medicine Iowa City, Iowa, U.S.A. The reviewer co-owns a spin-out company (Iowa Simulation Solutions) that has licensed patented simulator technology developed at the University of Iowa that might be considered a competitor to the training method to be studied.
REVIEW RETURNED	26-Mar-2020

GENERAL COMMENTS	General Comments: The multi-centre educational trial described in this manuscript addresses the important challenge of linking orthopaedic skills training to improvements in the operating theatre. The novelty of the approach isn't so much that a cadaveric training model is studied, but rather that efforts are made to evaluate if that training improves performance in surgery. This presents challenges that are ably addressed by the manuscript authors, although I would have appreciated a more balanced presentation of the strengths/weaknesses of the cadaveric training model being studied (see below). Otherwise, I think that this study will provide very interesting new information that lays a foundation for evaluating ongoing improvements in training methods. Specific Comments: Page 5, first 4 lines: It strikes me as appropriate to here introduce some concepts related to the feasibility of the proposed training methods. Has any consideration been given to challenges in scaling up of this training intervention to wider use? Is it indeed feasible to do this, given cost and limited availability of suitable cadaveric tissues (i.e., in this case specimens spanning from waist-to-toe-tip)? Page 8, lines 18-23: It strikes me that cadaveric tissues do have some limitations compared to live tissues that are unfortunately not here acknowledged. I can understand how the anatomic properties may be superior to other training alternatives, but it is not quite as clear to me that the tactile (haptic) properties are superior. It might be reasonable to provide a more balanced statement here. Page 17, first sentence: It would be helpful to have a more detailed description of what constitutes "normal clinical training 'on-the-job' according to the master-apprentice model." Is there assigned reading in advance of the case? Any didactic content that is delivered? At the very least it would be good to be able to assess how influential was merely being more exposed to thinking about the surgery as part of the cadaveric training. How might this be more closely controlled? Page 18, line 13: The sentence here should read "Hence each participant..." Please correct grammatical error. Page 21, lines 39-47: Has any consideration been given to how intervention by a supervising surgeon will be managed? It seems to me that in an educational setting, this is a legitimate possibility and that its nature would vary by surgeon and case difficulty. This latter point may come into play with the idiosyncrasies of different fracture patterns encountered that may be more or less difficult to reduce and fix. Furthermore, supervision by its nature implies that effort will be invested in getting an acceptable surgical result, granted perhaps at the expense of added time, and that this may limit your ability to discriminate between different surgical performances. Page 22, line 39/40: I'm not so sure that the cadaveric simulation training intervention is novel. Hasn't this played a role in skills
--

	training for at least decades, if not centuries? Please re-write to more precisely characterize this reality.
--	---

VERSION 1 – AUTHOR RESPONSE

Reviewer: 1

Reviewer Name

Amandus Gustafsson

Institution and Country

Copenhagen Academy for Medical Education and Simulation

Please state any competing interests or state 'None declared':

None declared

Please leave your comments for the authors below

Thank you for the opportunity to review your manuscript.

Investigating evidence of transfer of learning from the simulated setting to the clinical setting is important. Especially since the training modality of cadavers has been around for centuries, but as the authors point out, only have been investigated to a small degree compared with more novel modalities as virtual reality and computer assisted simulators. I find the protocol interesting, but have quite serious concerns chiefly between planned sample and outcomes.

Thank you for your thorough review. Having addressed your helpful suggestions we feel the strength of our protocol manuscript is much improved – many thanks. Our responses are detailed below each point raised.

Though there is several meta-analysis available on simulation-based training and the effect on patient outcomes that a sample size calculation could be based on, I agree with the authors that is reasonable with a pragmatic approach as training modality and outcomes vary hugely from previously published studies and this study. Also, the number of participants is aligned with similar studies. However, as the proposed study is planed, I'm seriously concerned for a type 2 error. The two main issues are 1) supervisor bias and 2) insufficient dose/exposure.

When conduction transfer studies from a simulated setting to the clinical setting for procedures of high complexity an invariable obstacle is the bias of the supervising surgeon as it would be hugely unethical to let novice surgeons operate on their own. We must assume that the senior surgeon will correct the trainee or even temporarily take over the operation to assure an optimal surgical result. This will in my opinion to quite a high degree reduce the effect, if any. This is true of both radiological outcome, but especially so for patient outcomes.

Re: 1) supervisor bias

Thankyou and we agree that accurately measuring supervisor input is challenging and a potential confounder in all real-world simulation transfer studies. In the UK surgical electronic logbook we have a coding system for the degree of supervisor input to an operation. We are only including procedures coded as 'supervised trainer scrubbed (S-TS)' and 'supervisor trainer unscrubbed (STU)'. These are defined as ST-S: 'The trainee performs the keys parts of the procedure as defined in the relevant PBA (procedure based assessment)' and for ST-U: The trainee performs the procedure from start to finish. More details on the supervision descriptors are available here <http://e1v1m1.co.uk/wp-content/uploads/2013/11/Supervision-codes-help-guide.pdf>

We will have access to PBA data on the procedures performed and, if required, this will give us further insight into precisely what parts of the procedure have been performed by the trainee. Instances of consultant/attending 'take-over' would be recorded in the free text domain of the PBA.

We have added this extra information to the manuscript under the 'Data Collection Plan' section. We will also acknowledge the challenge of supervisor bias when reporting our results.

Re: 2) risk of type 2 error

With regards to the ability to detect an effect of the training and risk of type 2 error, we have deliberately included a wide range of outcome measures as this is a highly exploratory study. As this is the first study attempting to detect real-world clinical impact of cadaveric training for open surgery, we do not know which, if any, of the proposed outcome measures will be responsive.

We believe the radiological outcome measures we have chosen will have sufficient resolution to detect small incremental gains in technical skill and are clinically relevant. There is an emerging body of literature to suggest that 'final product analysis' measures using post-operative x-rays satisfies many of the utility domains of effective assessment. Whilst there is clearly more validation work to be done around these, we believe they are worthy candidate outcome measures for this trial.

Two of our main stated objectives are to explore the feasibility of using post-operative x-rays to assess technical skill, and to attempt to define the real-world early surgical learning curves for the three procedures under study. We believe investigating and reporting on the feasibility of using these radiological and clinical outcomes to measure real-world early surgical learning curves will be a valuable addition to the translational simulation literature.

Your dose, in this case cadaver procedure training, is very small. You plan to let the intervention group train each procedure only once. Though this intervention may have an effect, I find it very unlikely that you will detect it with your sample size. To illustrate, I have recently published a study (Gustafsson A, Pedersen P, Rømer TB, Viberg B, Palm H, Konge L. Hip-fracture osteosynthesis training: exploring learning curves and setting proficiency standards. Acta Orthop 2019; 90(4): 348-53) on simulation-based training. Among others, we found that the median amounts of attempts the novices need to reach their learning plateau was 8 (range: 4-18).

Thankyou, we acknowledge that the training dose is small and we have now added this to the study limitations sections. This is a limitation facing almost all cadaveric training studies given the cost of the materials. In our study the participants are paired and although they will only perform each of the four procedures once as primary surgeon, they will be acting as assistant/scrub nurse for their partner and will be privy to the teaching and feedback given to them, therefore essentially doubling the training exposure. We think it is fair to assume there is considerable educational value in acting as assistant/scrub nurse at this junior level. We also have a generous faculty:delegate ratio of 1:2 so we anticipate the teaching during each cadaveric operation will be intensive and of high quality, considerably more so than in the real operating theatre.

We read your excellent recent paper with interest. It should be noted that we are not attempting to train participants to competence in the cadaveric simulation setting, and we hypothesise that the learning curve will continue in the real-world operating theatre during follow up. We are interested to see how the cadaveric training intervention might alter the subsequent learning trajectory as compared to the control group during the follow up period.

Also, your subjects are quite heterogeneous. You state that you will include trainees in their specialist training year 1 through 3. This is a huge difference in baseline competency and I would fear that the effect of the first year trainees will be washed out or be diminished by the third year trainees. It's not stated, but at the very least you will have to look into this problem during randomization as a skewness between the groups will seriously harm the interpretation of results.

We recognise that heterogeneity of participants is potentially problematic and to account for this we will collect participant baseline characteristics including number of completed months of T&O training (to the start of the study) and number of each of the procedures performed at baseline (stratified by supervision code). This will allow for clear comparison between the groups and for appropriate statistical adjustment to be made if we find evidence of skewness. We will also adjust for these variables in our learning curve analysis model. We have added a line in the Statistical Analysis Plan to make this clearer.

Based on your available sample size I would suggest that you readjust your focus from radiological and clinical outcomes measures to trainee behavioural outcomes with validated assessment tools, if you choose to continue to address comparing training to no training.

Thankyou we appreciate this suggestion. The extra value and novelty of our study is in attempting to provide real-world, patient based evidence of impact of cadaveric simulation training, and to report on the feasibility of doing this. We will not be able to demonstrate evidence of patient benefit or achieve the highest level of evidence of educational impact ('Kirkpatrick level 4') using workplace based behavioural assessments.

On the above points I would suggest you read Cook DA, Hatala R. Got power? A systematic review of sample size adequacy in health professions education research. *Adv in Health Sci Educ* 2015; 20: 73-83 and Cook DA, West CP. Reconsidering the Focus on "Outcomes Research" in Medical Education: A Cautionary Note. *Acad Med* 2013; 88: 162-67.

Thankyou for signposting this literature it is appreciated.

Specific comments:

Page 7, line 52-54: "where skills can be rapidly acquired" – do you have any evidence that skills can be acquired especially fast in a simulated setting as supposed to clinical practise?

We are not aware of any existing comparative studies that compare speed of learning in the simulator lab vs the clinical environment for open surgery. We are hypothesising here that the intensive nature of cadaveric simulation may lead to more rapid skill acquisition, and have amended the text to reflect the fact this is a hypothesis and not a statement of fact.

Page 7, line 54: "competency can be assessed before trainees are released into clinical practice." as I understand it, the study does not aim to assess the trainee's competencies in the simulated setting.

That is correct, we are not aiming to assess trainee competence in this study. We were referring to the value of simulation training in offering the opportunity to assess competence in a setting away from patients, on a conceptual level. We have removed this statement as agree it is misleading.

Page 7, line 56: "Simulation is also potentially a very effective way of training" I disagree. There's ample evidence that simulation-based training of procedural skills is the superior training modality.

We say potentially 'efficient', not 'effective'. We agree that the effectiveness of simulation training is well established. The efficiency however, particularly cost-efficiency, is less well understood.

Page 20, line 18: "The trial will end when data collection is complete." I cannot find any information in the manuscript as to when data collection is complete. Please state.

Apologies this was omitted in error. This has been revised to say ‘The trial will end when all the radiographic and clinical outcome data has been collected from the participating sites’.

Page 20, line 47: “Data collection plan” If possible, it would certainly improve the confidence of your results, if you could report how many of the investigated procedures the trainees had performed prior to study enrolment.

Thankyou we have added a statement to the Data Collection Plan which says ‘Data on the numbers of procedures performed by the participating surgeons at baseline will be collected’.

Reviewer: 2

Reviewer Name

Don Anderson

Institution and Country

Department of Orthopedics and Rehabilitation
University of Iowa Carver College of Medicine
Iowa City, Iowa, U.S.A.

Please state any competing interests or state ‘None declared’:
The reviewer co-owns a spin-out company (Iowa Simulation Solutions) that has licensed patented simulator technology developed at the University of Iowa that might be considered a competitor to the training method to be studied.

Please leave your comments for the authors below
General Comments:

The multi-centre educational trial described in this manuscript addresses the important challenge of linking orthopaedic skills training to improvements in the operating theatre. The novelty of the approach isn't so much that a cadaveric training model is studied, but rather that efforts are made to evaluate if that training improves performance in surgery. This presents challenges that are ably addressed by the manuscript authors, although I would have appreciated a more balanced presentation of the strengths/weaknesses of the cadaveric training model being studied (see below). Otherwise, I think that this study will provide very interesting new information that lays a foundation for evaluating ongoing improvements in training methods.

Thankyou for reviewing our manuscript and identifying areas for improvement. We have addressed your comments and responded in turn below.

Specific Comments:

Page 5, first 4 lines: It strikes me as appropriate to here introduce some concepts related to the feasibility of the proposed training methods. Has any consideration been given to challenges in

scaling up of this training intervention to wider use? Is it indeed feasible to do this, given cost and limited availability of suitable cadaveric tissues (i.e., in this case specimens spanning from waist-to-toe-tip)?

Thankyou we agree that feasibility is an important issue particularly given how expensive cadaveric training is, and that it relies on specialist facilities to deliver. We have added a line to the introduction section of the abstract saying 'The feasibility of delivering cadaveric training, use of radiographic and clinical outcome measures to assess impact and the challenges of upscaling provision will be explored'.

We have also added description of some of the challenges of providing cadaveric simulation training (ie cost, logisitics and infrastructure) in the introduction section of the manuscript.

Page 8, lines 18-23: It strikes me that cadaveric tissues do have some limitations compared to live tissues that are unfortunately not here acknowledged. I can understand how the anatomic properties may be superior to other training alternatives, but it is not quite as clear to me that the tactile (haptic) properties are superior. It might be reasonable to provide a more balanced statement here.

Thankyou. We have re-written this section to include a more balanced statement and removed reference to superior haptic properties. We have also added an acknowledgement of the fidelity limitations of cadaveric simulation (the main one being that cadaveric material does not bleed).

Page 17, first sentence: It would be helpful to have a more detailed description of what constitutes "normal clinical training 'on-the-job' according to the master-apprentice model." Is there assigned reading in advance of the case? Any didactic content that is delivered? At the very least it would be good to be able to assess how influential was merely being more exposed to thinking about the surgery as part of the cadaveric training. How might this be more closely controlled?

We have added further information to explain what is meant by standard residency training, and that this includes fortnightly didactic sessions that are delivered as a routine part of training.

Page 18, line 13: The sentence here should read "Hence each participant..." Please correct grammatical error.

Thankyou this has now been amended.

Page 21, lines 39-47: Has any consideration been given to how intervention by a supervising surgeon will be managed? It seems to me that in an educational setting, this is a legitimate possibility and that its nature would vary by surgeon and case difficulty. This latter point may come into play with the idiosyncrasies of different fracture patterns encountered that may be more or less difficult to reduce and fix. Furthermore, supervision by its nature implies that effort will be invested in getting an acceptable surgical result, granted perhaps at the expense of added time, and that this may limit your ability to discriminate between different surgical performances.

Thankyou. The issue of supervisor intervention/introduction of bias by the supervising surgeon was also raised by the other reviewer. We recognise that this is a common problem with translational simulation research. In response we have added an explanation that we will only be including procedures in the analysis which were coded as ‘supervised trainer scrubbed’ and ‘supervised trainer unscrubbed’ so we can be sure that the trainee actually performed the key parts (S-TS) or all (STU) of the procedure. We will also have access to procedure based assessments for these operations, which are completed routinely as part of training and which are not part of the study outcomes. If required, the PBAs will give further information as to precisely what steps of the procedure were performed by the participants and should document any instances of attending ‘take-over’. We have added this extra information to the manuscript under the ‘Data Collection Plan’ section.

With regards to case difficulty, we would expect that this would be balanced between the groups because of randomisation. We acknowledge this is not guaranteed and will be discussed as a limitation of the study when we report the results.

Page 22, line 39/40: I'm not so sure that the cadaveric simulation training intervention is novel. Hasn't this played a role in skills training for at least decades, if not centuries? Please re-write to more precisely characterize this reality.

We agree and have removed the term ‘novel’. It now reads ‘the cadaveric simulation training intervention is an experimental educational intervention’

VERSION 2 – REVIEW

REVIEWER	Amandus Gustafsson Copenhagen Academy for Medical Education and Simulation, Rigshospitalet, Denmark
REVIEW RETURNED	01-Jun-2020
GENERAL COMMENTS	The authors have addressed some of the reviewers concerns regarding skewness of prior experience by obtaining data on numbers of procedures and months of training at baseline. Also, information of the extent of supervision on the logged procedures has been added.

	This information will add clarity to the interpretation of the results, but it does not do much to improving the methodology of the study and my concern is still a type 2 error due to limited training exposure, supervisor bias and possible study arm skewness of prior surgical experience. Though the results of the study will be interesting if a type 2 error does not occur, I do not see how this protocol adds valuable new insights to the field of medical education.
REVIEWER	Don Anderson Department of Orthopedics and Rehabilitation University of Iowa Carver College of Medicine Iowa City, Iowa, U.S.A. The reviewer co-owns a spin-out company (Iowa Simulation Solutions) that has licensed patented simulator technology developed at the University of Iowa that might be considered a competitor to the training method to be studied.
REVIEW RETURNED	27-May-2020
GENERAL COMMENTS	The authors are to be congratulated on successfully responding to all of my prior critique. I have no further concerns.

VERSION 2 – AUTHOR RESPONSE

Reviewer 1 comment;

my concern is still a type 2 error due to limited training exposure, supervisor bias and possible study arm skewness of prior surgical experience

Our response;

We thank reviewer 1 for their comments.

A type 2 error (a false negative result) is a known risk in all trials, and impossible to fully mitigate. We accept the training dose is relatively 'small' in our study, being a single cadaveric training course. This intervention is however very intense in educational terms, as it takes place over 2 days and involves the trainees performing multiple operations in their entirety with one-to-one supervision and feedback from attending faculty surgeons. We believe part of the value of cadaveric training is this high learning yield from a single intervention.

Regarding supervisor bias, the course faculty are all from a single hospital (one of the nine study sites) and therefore only trainees who are working in the sentinel site will be at any potential risk of supervisor bias. These trainees will be balanced between the two study arms. Although it is not

realistically possible to formally blind supervising surgeons in the nine hospitals to trainee group allocation, we believe the real life chance of supervisor bias to be low.

Regarding study arm skewness of prior surgical experience, we assume this will be balanced between the two groups by randomisation. We will also adjust for this in our analysis “**Linear mixed effects models** will be fitted to allow for within-surgeon correlation between repeated observations (surgeon clustering as a random effect), and to adjust for important co-variables such as patient condition, age and **surgeon experience**. These will be summarised by plotting individual learning curves, and then modelled to estimate the overall learning curves for the two arms of the study.”